# Generation of 3D Realistic Soil Particles with Metaball Descriptor

**Yifeng Zhao**
Zhejiang University & Westlake University

**Jinxin Liu**
Westlake University

**Xiangbo Gao**
Westlake University

**Pei Zhang**
Westlake University

**Stan Z. Li**
Westlake University

**Sergio Torres**
Westlake University [*]

## Abstract

The accurate representation of soil particle morphology is crucial for understanding its granular characteristics and assembly responses. However, incorporating realistic and diverse particle morphologies into modeling presents challenges, often requiring time-consuming and expensive X-ray Computed Tomography (XRCT). This has resulted in a prevalent issues in modeling: morphological particle generation. On this topic, we introduce the Metaball Variational Autoencoder. This method leverages deep neural networks to generate new 3D particles in the form of Metaballs while preserving essential morphological features from the parental particles. Furthermore, the method allows for shape control through an arithmetic pattern, enabling the generation of particles with specific shapes. We validate the generation fidelity by comparing the morphologies and shape-feature distributions of the generated particles with the parental data. Additionally, we provide examples to demonstrate the controllability of the generated shapes. By integrating these methods into the Metaball-based simulation framework proposed by the authors previously, we enable the incorporation of real particle shapes into simulations. This could facilitate the simulation of a large number of soil particles with varying shapes and behaviors, providing valuable insights into the properties and behavior of actual soil particles.

## 1 Introduction

On elucidating the impact of particle shape on the granular soil, simulation schemes based on micromechanical models, especially the discrete element method (DEM) [1], have prevailed in the past few decades. In original DEM, the particle is simplified as circles or spheres, which is hard to reflect the impact of shape, e.g. the resistance to rolling. Under this context, the key issue is how to fully reconstruct the morphology of granular matters in DEM simulations.

With the inspiring development of X-ray Computed Tomography (XRCT) and computer vision techniques, opportunities are provided to bring more accurate and sophisticated shape features into DEM simulations. Various reconstruction methods, which are called shape descriptors in this paper, are developed to compress realistic particle morphologies into a uniform mathematical representation for simulations. A good example is the Fourier descriptor, which is developed to capture particle shapes based on the average normalized Fourier spectrum of main contours from the targeted particle [2, 3]. Through similar pattern, the Spherical-Harmonic (SH) descriptor can also be used to capture various shapes features [4–7]. However, these methods are limited to tackling star-like particles, of which all line segments between particle-center and particle-surface points are located

---

[*]Corresponding author: s.Torres@westlake.edu.cn

within the particle body [6]. Many particles, such as lunar soils and concave sand, do not follow such constraints. Recently, the Metaball descriptor was introduced by [8] to reconstruct non-spherical particle shapes. With proper function form, the contact detection of it can be tackled at a low cost, which enables a more efficient simulation framework [9]. Such framework is further coupled with Lattice Boltzmann Method for simulations of more complicated physical processes in fluid-particle systems [10].

However, although XRCT can be used to scan all involved particles in the application, particle scanning and image processing can be economy-costly and time-consuming. In practical engineering applications, it is typical that only a small fraction of particles can be analyzed due to limited resources, as reported in [11–13]. Direct simulation with them will suffer from repetitive particle morphologies. This makes it necessary to generate realistic particles with coessential morphological features. With the development of the aforementioned shape descriptors, many attempts have been inspired to tackle generation tasks. Among them, the SH-based technique is a popular choice [14, 15]. It first incorporates geometric features into specific SH coefficients. Then, different algorithms like random field [16], fractal dimension [17], Nataf transformation [11], and principal component analysis [18, 13] are applied on the distilled SH coefficient to add small variances following the morphological pattern of parental particles for generation. But the above schemes suffer from some problems, including 1) underfitting and overfitting problems on shape-feature distributions of the generated particles, e.g. the distributions of surface area and volume [12, 18]. 2) Hard to obtain particles with specific morphological features [19, 14], e.g. generating non star-like particles with angled features. 3) Involving complex mixture models, which treat particle generation as a high-dimensional, multi-parameter estimation problem [20, 21]. Such a method can achieve an advance in performance, but still has a reliance on computational and human resources, which can not be obtained easily by all individuals or institutions [12]. 4) Requiring bridging or transformation into other descriptors before practical simulations, which often results in a trade-off between accuracy and efficiency [22, 23]. Thus, a framework with a more flexible shape descriptor, to tackle the above problems, is needed.

On the above dilemmas, this paper presents Metaball-based two-step solutions. We first utilize a Metaball-Imaging (MI) algorithm to capture complex particle morphologies from XRCT images with the Metaball descriptor. Then, we develop a Metaball-based Variational Autoencoder (MetaballVAE). It can learn from the XRCT image of targeted grains and generate random Metaball-based particles retaining major morphological features from a regularized latent space, where complex calculations are converted into one-step solutions. Note that MetaballVAE can learn not only the geometric features of the single particle but also feature distributions of the particle group. The regularized latent space also makes it possible to modify particle morphologies in an arithmetic pattern, allowing for obtaining particles with specific shapes. Examined with two groups of particles different in sample number and geometric characteristics, the proposed generation algorithm has proved to be robust and effective.

## 2   Our Method

The variational autoencoder (VAE) [24] is a neural-network based generative model. The framework of it is to first learn the distribution $P$ of the targeted data $\boldsymbol{x}$ and then generate through sampling with some unobserved variable $\boldsymbol{z}$, which is called the latent variable. And the collection of them is named the latent space. In implementation, the learning of $P(\boldsymbol{x})$ is carried out with an assumed distribution $\int Q(\boldsymbol{z} \mid \boldsymbol{x})Q(\boldsymbol{x})d\boldsymbol{z}$ ($Q$ is the assumed distribution of two parts. Since these two parts are implemented in one neural network system, they share the same notation) , which is in the form of neural network. This distribution corresponds to two important components of VAE: the encoder $\boldsymbol{z} = E(\boldsymbol{x})$(For $Q(\boldsymbol{z} \mid \boldsymbol{x})$) and decoder $\bar{\boldsymbol{x}} = D(\boldsymbol{z})$(For $Q(\boldsymbol{x})$), where $\boldsymbol{x}$ represents the input, $\bar{\boldsymbol{x}}$ for the generated(reconstructed). For particle generation, $\boldsymbol{x}$ and $\bar{\boldsymbol{x}}$ refer to the shape representation, for example, the Metaball descriptor $\boldsymbol{M}$ or XRCT images. The encoder and decoder consist of the major steps in VAE: encoding and decoding. In encoding, the input shape representation $\boldsymbol{x}$ is compressed and mapped into the latent variable $\boldsymbol{z}$, a multidimensional shape-representation tensor. Then $\boldsymbol{z}$ is decoded to reconstruct the input particle $\bar{\boldsymbol{x}}$. Through minimizing the difference between $\boldsymbol{x}$ and $\bar{\boldsymbol{x}}$, morphologies and shape-feature distribution of input particles can be learned effectively (The theory behind this is briefly stated in Appendix A). Then, the trained decoder $\bar{\boldsymbol{x}} = D(\boldsymbol{z})$ can be applied to generate particles by inputting random $\boldsymbol{z}$. Assisted by the powerful learning ability of neural network,

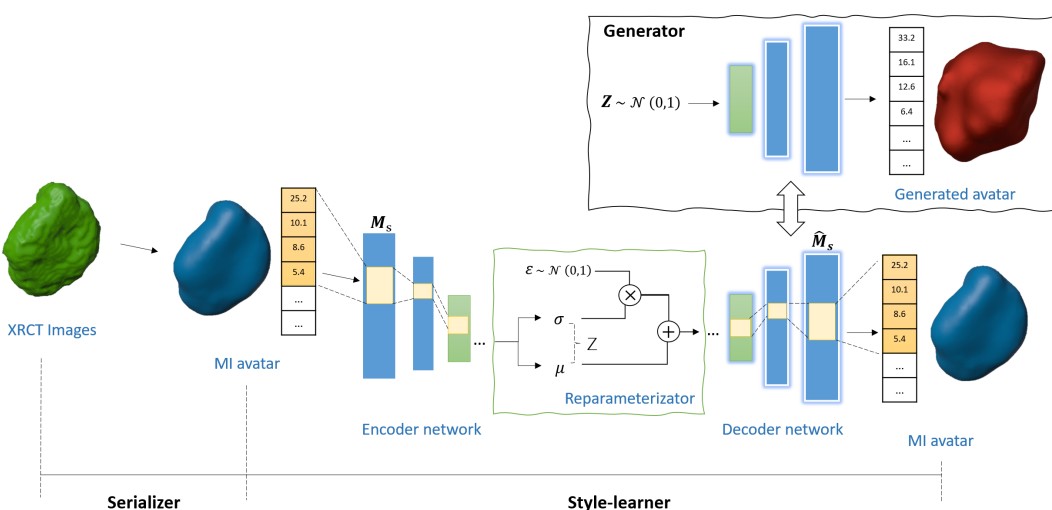

Figure 1: MetaballVAE for generation of complex-shaped particles. Serilizer: interpreting and transforming XRCT images of granular matters. Style-learner: analysing the serilized Metaball descriptor and learning the morphological characteristics and distribution. Generator: generating style-similar granular media in Metaball form

VAE can inference new particles, which are not included in the training set but maintain coessential morphological features and distributions with the parental particles. This make it have the potential to provide a more practical solution to the particle generation task than previous studies.

It is worth noting that latent variables $z$ in VAE are regularized (chosen to be multivariable normal distribution) to encourage similar input samples compress at closer positions in the latent space. This property allows the model to learn a more flexible and general distribution, rather than simply adapting to the specific patterns present in the training data. As a result, generation by sampling from sperate, regularized $z$ can help to avoid overfitting and underfitting problems on the shape-feature distributions of generated particles [21, 12]. More importantly, it enables high-level controls on the generated particle morphologies [25].

Based on VAE, we propose a Metaball-based particle generation framework called Metaball Variational Autoencoder (MetaballVAE). It can resolve the correlation between XRCT images and morphologies of input particles with a regularised latent space, where complex computations are transformed into one-step solutions, reducing variance in generated models and improving control over the generation process. This model requires no prior knowledge (e.g. particle shape-feature distributions), but only XRCT images of the target particles to generate non-existent style-similar avatars, particles in the form of metaballs, which can be used directly in simulations. Note that style learning is not limited to morphology, but also to shape-feature distributions.

The MetaballVAE consists of three major parts as illustrated in Figure 1: serializer, style-learner and generator. The serializer interprets and transforms XRCT images of target granular-particles into Metaball descriptors. Then, the style-leaner analyses those distilled descriptors, capture major shape characteristics, conducts inference on feature distribution and devise style-similiar avatars. In the end, the generator outputs designed style-similar, Metaball-based avatars.

## 2.1 The serializer

The serializer is designed to abstract the particle morphology, extract shape-feature distributions and code them structurally. It can significantly reduce the dimension of XRCT images while keeping all the vital morphological information for generation. The reasons for implementing it are two-fold. On the one hand, structured data are more suitable for generation tasks [26], which can improve the generation quality. On the other hand, this enables a direct generation of particles in Metaball function form, which can be put into simulations without bridging or transformation, avoiding unnecessary information loss and computational cost.

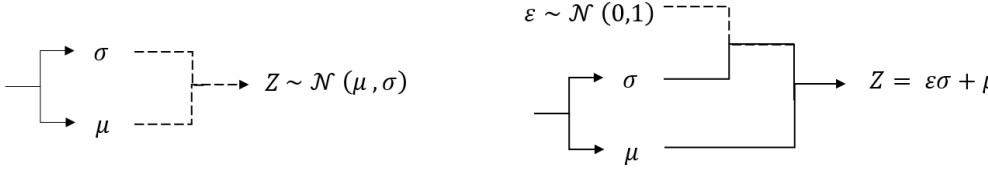

(a) Direct sampling  (b) Reparametrized trick

Figure 2: Visualizations of the direct sampling and reparameterization trick. The solid line stands for a relationship capable of backpropagation. The dashed line represents a relationship where backpropagation can not be carried out.

In this paper, the serialization is accomplished with the Metaball-Imaging technique as introduced in Appendix A. The serilized particle is in MI avatar form, which is noted as $M_s$.

## 2.2 The style-learner

The style-learner is a modification of the aforementioned VAE. It can digest the structured data $S$ and learn how to generate particles. Main components involved are: encoder, decoder, reparameterizator, loss function and distribution annealer.

**Encoder** and **Decoder** are multi-layer perceptions, which are connected in a bottle-neck form as shown in Figure 1. They are implemented to approximate the real distribution $P(x)$ as a learnable, assumed distribution $\int Q(z \mid x)Q(x)dz$. On the topic of particle generation, $x$ refers to the $M_s$. The encoder takes serialized particles $M_s$ as input and outputs parameters($\mu$ - the mean, $\sigma$ - the standard deviation) of the corresponding latent variable $z$, mapping morphologies and shape-feature distributions of particles into a regularized latent space. On the contrary, the decoder interprets $z$ to restore $\hat{M_s}$, reconstructing particle morphologies and shape-feature distributions from that regularized space.

**Reparameterizator** locates halfway between the encoder and decoder. It is designed to regularize the latent space by creating a map between the encoded information and a normal distribution. Instead of direct sampling (Fig, 2, a), a more deterministic pattern is utilized:

$$z = \mu + \sigma \odot \epsilon \tag{1}$$

where the $\epsilon$ is the assumed normal distribution. This enables continuous gradient calculation on the mapping relationship, making MetaballVAE a learnable system.

**The loss function** is the global optimization objective for MetaballVAE:

$$L(M_s) = \underbrace{\frac{1}{d}\sum_{k=1}^{d}\|M_s - \hat{M_s}\|^2}_{\text{Reconstruction Item}} + \underbrace{\frac{1}{2}\sum_{k=1}^{d}\left(\mu_{(k)}^2(M_s) + \sigma_{(k)}^2(M_s) - \ln\sigma_{(k)}^2(M_s) - 1\right)}_{\text{Distribution Item}} \tag{2}$$

where $d$ is the dimension of $M_s$. This function is modified from the original VAE theory based on the particle generation problem. The deduction of it is stated in Appendix B. It consists of two items: the distribution item and reconstruction item. The distribution Item measures the difference between real and learned distributions of particle morphologies. The Reconstruction Item evaluates the quality of learned morphological characteristics. The combination of them forces MetaballVAE to learn not only morphologies but also shape-feature distributions of the input particles.

**The distribution annealer** is proposed to tackle the training challenge of VAE. A well-trained model possesses a relatively small reconstruction item and a non-zero distribution item. However, most direct training will yield a model with a zero distribution item. Such tendency in learning is caused by the sensitivity of decoder to variation introduced by the mapping process of reparameterizator. This makes the decoder ignore the latent variable provided by the encoder and output the average

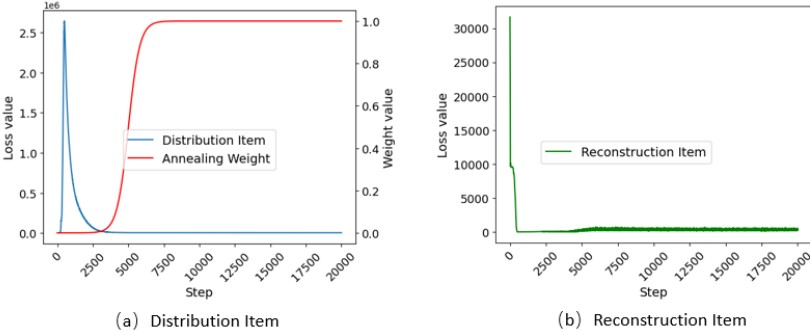

(a) Distribution Item                    (b) Reconstruction Item

Figure 3: The impact of the distribution annealer on the loss value of different items in the training of cobblestone dataset

optimal with distribution item equal to zero. For this reason, the distribution annealer is implemented by adding a weight to the distribution item of the loss function(Eq. 2). This weight starts from zero, where the weighted loss function equals the reconstruction loss. Then, the weight is increased gradually to one, where this weighted function satisfies the true loss function definition. With such a process, the model will be forced to use the learned latent space to achieve good likelihood in prediction.

Figure 3 is an example of the distribution annealer in the training of the cobblesteone dataset during the first 20k steps. It can be observed that the distribution item first spikes as the reconstruction item drops significantly, where the model is encoding shape features into the latent space cheaply. Then, the distribution item starts to decrease rapidly as more attention is paid to the divergence penalty. Correspondingly, the decrease of reconstruction item slows down. Finally, the distribution item gradually converges and the reconstruction item enters fluctuation, where more morphological information is compressed into the model.

## 2.3   The generator

Before formal generation, the decoder of style-learner should be well-trained. The generation task requires the trained decoder and a normal distribution $N(0,1)$, which represents the regularized latent space. A typical generation process is illustrated in Figure 1. The latent variable $z$ is sampled from a normal distribution, serving as the input matrix to the decoder. Then, the decoder can devise style-similar particles unseen in the training dataset. The distribution of shape features can be well reconstructed when the number of generated particles is large enough. It is worth noting that the generated particle is in the form of Metaball descriptor, which can be applied directly into simulations.

# 3   Experiments

## 3.1   The setting of serializer and style-learner

In this evaluation, we apply the following hyper-parameter setting. In serializer, the control point number, n, is set to be 40 for both cobblestone and Ottawa sand samples. The learning rate for the gradient update is set to be 0.001. In style-learner, the encoder is a 4 layer full-connected network with leaky ReLU activation function. The size of it is: $160{\times}1024{\times}512{\times}256{\times}128$. The decoder is also a 4 layer full-connected network with leaky ReLU activation function yet in reverse form. The reparameterizator is set to be one fully connected layer with size 128. The above networks are trained by Adam [22] with learning rate $\eta = 0.0001$.

Since the setting of hyper-parameters is not a focus of this paper, how to obtain them is not included here for the sake of brevity. A detailed procedure can be referenced in [12].

## 3.2   Dataset and Metrics

Previous studies on particle generation are often carried out on hundreds of thousands of samples [12, 20, 25]. However, particle reconstruction with XRCT requires considerable time and computational

resources. In actual engineering, it is very often to have only a dozen scanned particles. Therefore, it's crucial to evaluate the performance of algorithms on smaller datasets. To this end, we tested the effectiveness of the MetaballVAE model on four distinct sets of XRCT data, which included particles of different types and sizes: 290, 100, and 10 samples of Ottawa sands, as well as 20 cobblestones. For better learning performance, data augmentation are implemented on training datasets, where slightly modified synthetic data is introduced based on the real one. Here, particle rotating and parameter shuffling are implemented. Particle rotating is a popular strategy based on rotational-invariant property. For example, Shi et al. [12] applied nine rotations to each particle and enlarged his dataset by ten times in a particle generation task, which effectively enhanced the model performance. Parameter shuffling means random recombination of $\{k_i, \boldsymbol{x_i}\}$ in the serialized particle $\boldsymbol{M_s}$. This is because the sequence change of control spheres will not modify the corresponding Metaball model. Such processing can effectively avoid the overfitting problem and enhance convergence performance.

Accurate evaluation is a challenge for particle generation tasks. Apart from the rationality of particle shape, another important content of evaluation is the quantitative difference between parents and clones. We select seven shape factors for evaluation: surface area $A$, volume $V$, Corey Shape Factor $CSF$, nominal diameter $D_n$, surface-equivalent-sphere diameter $D_s$, sphericity $\phi$ and circularity $C$.

The Corey Shape Factor $CSF$[27] reveals the dimension feature of the studied particle, as given by:

$$\text{CSF} = \frac{L_s}{\sqrt{L_i L_l}} \tag{3}$$

where $L_s$, $L_i$ and $L_l$ are the shortest, intermediate and longest axis lengths of particles.

The nominal diameter $D_n$ and surface-equivalent-sphere diameter $D_s$ are two widely used parameters[28, 29]. The $D_n$ is defined as the diameter of the volume-equivalent sphere. And the $D_s$ takes the following form:

$$D_s = \sqrt{\frac{4A_p}{\pi}} \tag{4}$$

where $A_p$ = the maximum projected area of the particle. Here, they are combined as $D_{ns} = D_n/D_s$ to form a dimensionless quantity.

The sphericity $\phi$ [30] is the measure of similarity between the studied particle and the sphere, which is defined as:

$$\phi = \frac{A_{ve}}{A} \tag{5}$$

where $A_{ve}$ = the surface area of the volume-equivalent sphere to the studied particle; $A$ = the surface area of the studied particle.

Another frequently used metric is the circularity $C$ [28], which evaluates the roundness of non-spherical particle:

$$C = \frac{\pi D_s}{P_p} \tag{6}$$

where $P_p$ is the the perimeter of the particle's projected-area.

### 3.3 Particle Generation

In validation, the training datasets are denoted as "Parents". And 1000 particle avatar are devised by the generator, denoted as "Clones". The number of generated particles is set to larger for better evaluation on the morphological distribution.

Figure 11 and Figure 12 displays several cloned examples of each type. It can be concluded that these clones exhibit reasonable shape features of both Ottawa sands (angled features) and cobblestones (round features). Note that these particles are in Metaball function form and the meshes are only for visualization.

To further exam the regeneration ability of the proposed method on shape-feature distributions, probability density functions (PDF) of selected metrics are calculated on both parents and their clones. A satisfying match can be observed (See details in Appendix B.)

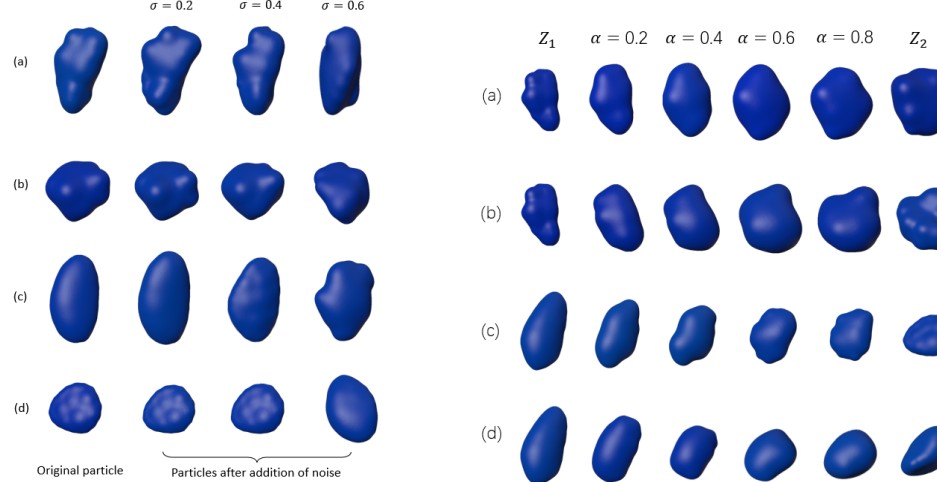

Figure 4: Comparison of generated particles before and after addition of Gaussian noise in the latent variables. (a) and (b) are Ottawa sand examples. (c) and (d) are cobblestone examples. The variance of Gaussian noise is noted as $\sigma$.

Figure 5: Comparison of generated particles of interpolated latent variables. (a) and (b) are cobblestone examples of the same $z1$. (c) and (d) are Ottawa sand examples of the same $z_1$. The interpolated coefficient is noted as $\alpha$.

## 3.4 Latent space arithematic

A well-trained MetaballVAE can achieve integration of discrete training samples into continuous, regularized latent space, where the number of generated particles can be infinite. Such regularized latent space avoids overfitting and underfitting problems in generation, ensuring the rationality of generated morphologies [12]. More importantly, it also realizes a certain control on the generated geometric feature [31].

Figure 4 illustrates that the addition of Gaussian random noises $\delta$ into the latent variable $z$ can introduce morphological changes of different degrees into generated avatars. Here, $\delta$ is set to have the same dimension of $z$, with zero mean value and different variance $\sigma$. Then, $\delta$s are added to a randomly selected latent variable $z$ to produce modified ones $z + \delta$. Finally, these latent variables are fed into the generator as inputs. From the corresponding generated results, it can be observed that the addition of $\delta$s with small $\sigma$ can slightly adjust the particle morphologies. As the increase in $\sigma$, the degree of modification becomes larger, resulting in less similar avatars to the original one. This is from the property of regularized latent space. The addition of $\delta$s can create new latent variable adjacent to the original one, while the $\sigma$ of $\delta$s decides distances between them in the latent space. Since the latent space is regularized, such adjacent relationships control the shape similarity in generated avatars. This phenomenon is also observed in generating digital sand particles with VAE [12]. It can be very useful when particles in certain morphologies are needed in simulations. We can first select the template avatars and then add $\delta$s of small magnitude into its latent variables. In this way, slightly modified avatars can be generated, avoiding repetitive particle morphologies in simulation.

Figure 5 indicates that interpolation between latent variables can produce smooth shape transitions in corresponding generated avatars. In these examples, two latent variable $z_1$ and $z_2$ are randomly selected to create interpolations $z_1 + \alpha(z_2 - z_1)$. Then, these latent variables are fed as input to the generator. It is clear that as the increase of $\alpha$, those interpolated avatars gradually transform from $z_1$ avatar to $z_2$ avatar. Note that such a change occurs simultaneously in multiple characteristics including shape, volume and surface area. This is also result of the regularized latent space. Those interpolated variables possess adjacent locations in the latent space with $z_1$ and $z_2$, resulting in avatars of similar shape. With change in the location of latent variables, the generated avatars show smooth modification in shape from $z_1$ to $z_2$ avatars. This phenomenon can be applied to obtain avatars of combined features. We can first select two template avatars of specific morphologies then do interpolation between them.

Figure 6 shows that the avatar shape can be modified by applying addition or subtraction in the latent space. Here, arithmetic operations are implemented on latent variables, $z_1$ and $z_2$, corresponding to avatars of distinct shapes. Under such operation, specific shape features can be added or removed from the generated avatar of $z_3$. This also results from regularized latent space. The proper mapping between latent variables and morphological features provides a powerful method to modify the shape. Through proper arithmetic operation, avatars of specific features can be generated according to need.

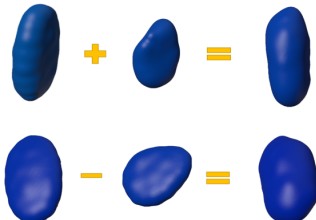

Figure 6: Manipulation of particle shape through addition and subtraction in latent variables. Top: adding angled features into a smooth cobblestone. Bottom: removing flattening features from a thin cobblestone.

## 4    Conclusion

We present a variational-autoencoder (VAE) based particle generation algorithm called MetaballVAE. It can generate style-similar particles in Metaball function form with XRCT images of parental particles. The MetaballVAE was evaluated through a comparison of two groups of particles with different sizes. It was found that the parental and cloned particles exhibited good agreement in terms of their morphologies and shape-feature distributions. These results provide evidence that MetaballVAE is a reliable and practical tool for characterizing particles with varying sizes and morphologies. The regularized latent space of MetaballVAE allows for control over the generation process. Particles with specific morphologies can be generated through arithmetic operations on the latent space. This feature makes MetaballVAE a versatile and useful tool for generating particles with desired characteristics. With previously developed metaball-based simulation frameworks that have proven to be a powerful tool in comprehending the intricacies of fluid-particle systems involving realistic soil particles [10, 8]. With the addition of reconstruction and generation methods proposed in this paper, the integration of these tools has the potential to unveil new insights into soil mechanics and provide valuable information for a wide range of applications, including soil erosion modeling, soil contamination analysis, and soil moisture modeling.

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

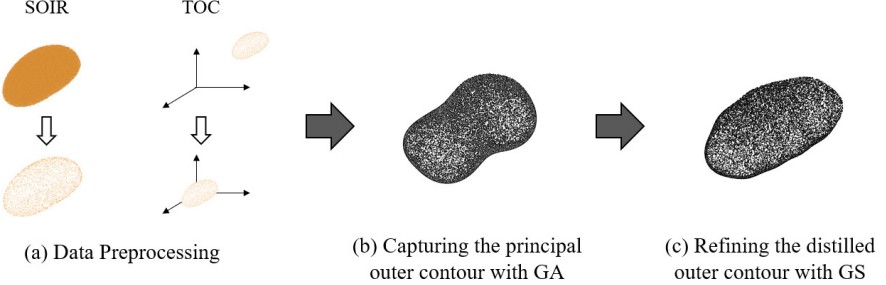

SOIR    TOC

(a) Data Preprocessing     (b) Capturing the principal     (c) Refining the distilled
                               outer contour with GA            outer contour with GS

Figure 7: The algorithm flow of the Genetic-based Gradient Search algorithm.

## Appendix A. The Metaball-Imaging based on genetic algorithm

Here we design a Metaball-Imaging (MI) algorithm to to transform the XRCT image of irregular-shaped particle into an explicit, Metaball-function based mathematical representation, which is called *avatar* in this paper. This task can be simplified into an optimization problem, searching for the set of parameters $\{\hat{k}_i, \hat{\boldsymbol{x}_i}\}$ for a Metaball model which can minimize the following function:

$$\arg\min_{\hat{k}_i, \hat{\boldsymbol{x}_i}} \left( \sum_{i=1}^{n_1} \sum_{j=1}^{n_2} \frac{\hat{k}_i}{\left(\boldsymbol{x}_j - \hat{\boldsymbol{x}_i}\right)^2} - 1 \right)^2 \tag{7}$$

where $\{\hat{k}_i, \hat{\boldsymbol{x}_i}\}$ is the parameter set of the targeted Metaball model; $\boldsymbol{x}_j$ stands for the input, coordinates of hull points from XRCT; $n_1$ and $n_2$ refer to the number of control points and input samples seperately.

The proposed algorithm involve three steps as shown in Figure 7. (1) Data preprocessing of the XRCT cloud points (Figure 7 a). (2)Capturing the principal outer contour with genetic algorithm (Figure 7 b). And (3) Refining the distilled outer contour with gradient search (Figure 7 c).

### .1   Data preprocessing of the XRCT image

This preprocessing consists of two operations, the specification of interested regions (SOIR) and the transformation of coordinates (TOC). SOIR is to specify those points on outer contour from the XRCT result. This is because the fitting of the Metaball only requires the point hull. TOC is implemented to translate the specified region of interest into the coordinate system centered at the origin. Such an operation can avoid abnormal fitted parameters caused by XRCT coordinates. The point hull obtained through SOIR and TOC is noted as $\boldsymbol{H} = \{Hx_i, Hy_i, Hz_i\}$ ($i \in [1, m]$, where m represents the number of hull points).

### .2   Capturing the principal outer contour with genetic algorithm

Considering the superiority of GA in global search, the ability to locate the range of the optimal accurately, it is used to capture the rough outer contour, i.e. a sufficiently good solution for Eq. 7.

In the capture of principal outer contour by GA, five segments are included: *population initialization*, *mutation*, *crossover*, *evaluation* and *selection*. Generations of these segments will be carried out as shown in Algorithm 1.

*Population Initialization*: This process refers to the initialization of N individuals as a population. Each individual represents a possible parameter set $\boldsymbol{M} = \{M_{k_i}, M_{x_i}, M_{y_i}, M_{z_i}\}$ ($i \in [1, n]$) to the fitted Metaball model (See Figure 8). An individual consists of a series of strings defined as Genes, standing for a certain parameter in the set. The number of genes in each initialization is set to be a constant.

In this segment, all individuals are randomized with control points inside targeted point hull $H$. Such an operation is to satisfy the geometric constraint of the Metaball. The judgment of whether a point

**Algorithm 1** The Genetic Algorithm for capture of the principal outer contour

**Input:** the preprocessed point hull $\boldsymbol{H}$, the number of generations $E^{ga}$, the number of individuals in the population $N_I$, the number of genes in each individual $N_G$, the mutation coefficient $C_m$, the crossover coefficient $C_c$.

**Output:** the Metaball model of the principal outer contour $\boldsymbol{M}^{ga}$.

1: **Initialization -** $N_I$ individuals are randomly initialized with a string of $N_G$ genes. The control points in each individual are set to be inside $\boldsymbol{H}$;
2: **for** $i = 1, 2, ...,$ to $E^{ga}$ **do**
3:     **Mutation -** For each indivudial, performing muatation with coefficient $C_m$;
4:     **Crossover -** For random pair of individuals, performing crossover with coefficient $C_c$;
5:     **Evaluation -** For each individual, calculating the fitness score;
6:     **Selection -** Selecting the fittest $N_I$ individuals for next generations;
7: **end for**
8: **Return:** The fittest individual in the population $\boldsymbol{M}^{ga}$.

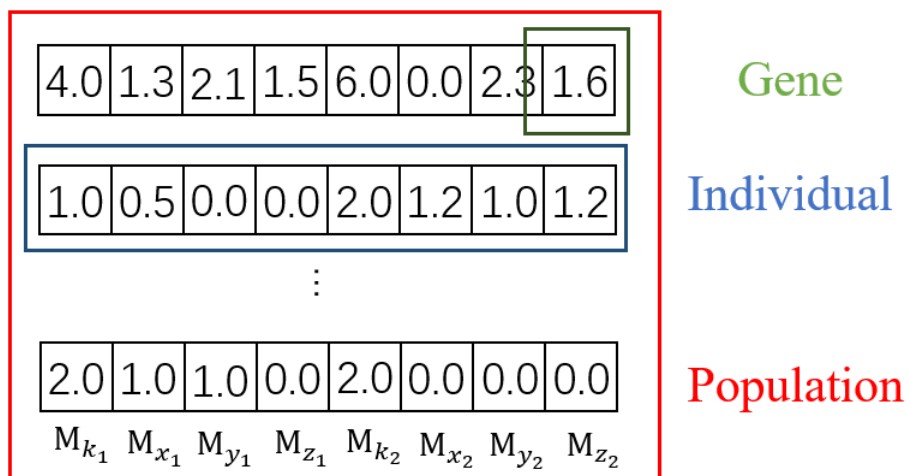

Figure 8: Population, individual and gene.

$P$ is inside $H$ is completed by linear programming. This problem is defined as the following:

$$
\begin{aligned}
\underset{A}{\text{Minimize}} \quad & \boldsymbol{C}\boldsymbol{A} \\
\text{Subject to} \quad & \boldsymbol{H}^T \boldsymbol{A} = \boldsymbol{N}^T
\end{aligned}
\tag{8}
$$

Where $\boldsymbol{C} = \underbrace{[1, 1, \cdots 1]}_{m}$; $\boldsymbol{A} = \underbrace{[a_1, a_2, \cdots a_m]}_{m}^T$; $\boldsymbol{H}^T = \begin{vmatrix} Hx_1, & Hx_2, & \ldots & Hx_m \\ Hy_1, & Hy_2, & \ldots & Hy_m \\ Hz_1, & Hz_2, & \cdots & Hz_m \\ 1, & 1, & \ldots & 1 \end{vmatrix}$,

$\{Hx_m, Hy_m, Hz_m\}$ is the coordinate of points from the preprocessed hull $\boldsymbol{H}$; $\boldsymbol{N} = [P_x, P_y, P_z, 1]^T$, $(P_x, P_y, P_z)$ is the coordinate of the studied point P.

If the point $P$ is inside the studied point hull $\boldsymbol{H}$, there will be a solution A to Eq. 8, satisfying:

$$
a_1 + a_2 + ... + a_m = 1, a_i > 0
\tag{9}
$$

*Mutation and Crossover*: Mutation and Crossover are strategies to produce offsprings. They are the most vital segments in GA and the key to finding the optimal solution.

Mutation refers to random change in the value of genes with a probability $C_m$, which stands for the change probability (Figure 9 I). This strategy is designed to control the exploration breadth and convergence rate. Crossover means the exchange of genes between two different individuals with a coefficient $C_c$, which determines the crossover point (Figure 9 II). This strategy is dedicated to

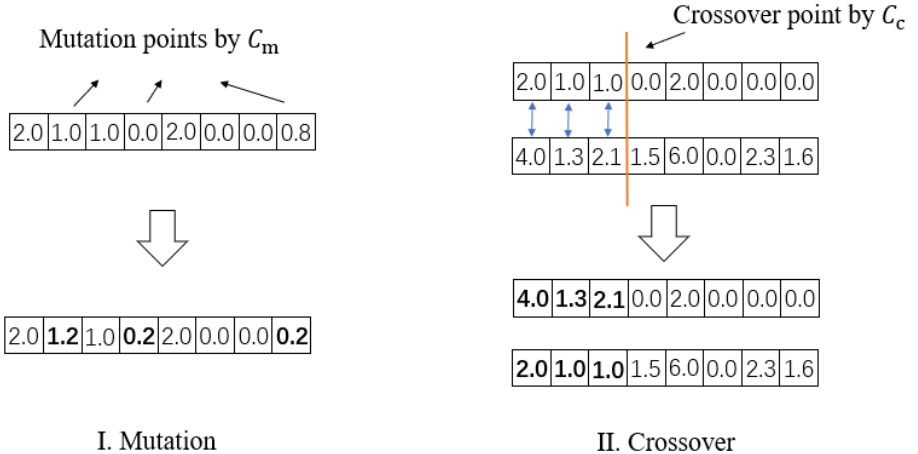

Figure 9: Mutation and Crossover.

filtering out better genes and promote positive diversity. A proper setting of $C_m$ and $C_c$ can maintain population diversity and prevent overfitting problems.

*Evaluation*: Evaluation involves the concept of fitness functions. It gives a fitness score, which determines the ability of this individual to compete with others in the population, to each individual. Here, the fitness function is defined as the following:

$$F(\boldsymbol{M}) = \sum_{i=1}^{m} (f_{H_i}^l(M) - 1)^2 \tag{10}$$

where $f_{H_i}^l(M) = \sum_{j=1}^{n} \frac{M_{k_j}}{\left(H_{x_i} - M_{x_j}\right)^2 + \left(H_{y_i} - M_{y_j}\right)^2 + \left(H_{z_i} - M_{z_j}\right)^2}$, $H_i$ stands for a control point in the point hull $\boldsymbol{H}$.

*Selection*: Selection is based on the fitness score calculated in Evaluation. It is implemented to filter the fittest N individuals and pass them to the next generation. It is generally performed after mutation, crossover and evaluation. This makes individuals with higher fitness more likely to survive and reproduce.

### .3 Gradient Search for refinement of outer contour

In the refinement of outer contour by GS, two segments are included in GS: *Gradient Descent* and *the anomaly detection*. The flowchart of GS is detailed in Algorithm 2.

---

**Algorithm 2** The Gradient Search for refinement of outer contour

---

**Input:** the preprocessed point hull $H$, the number of generations $E^{gs}$, the learning rate $\eta$, the Metaball model of the principal outer contour $\boldsymbol{M}^{ga}$.

**Output:** the metaball model for the refined outer contour $\boldsymbol{M}^{gs}$.

1: $\boldsymbol{M}^{ga}$ is taken as the inital parameter $\boldsymbol{M}_0$;
2: **for** $i = 1, 2, ...,$ to $E^{gs}$ **do**                                   ▷ **1st Gradient Descent**
3:     $\boldsymbol{M}_0 \leftarrow \boldsymbol{M}_0 - \eta \cdot \nabla_{\boldsymbol{M}_0} L(\boldsymbol{M}_0)$;
4: **end for**
5: **Anomaly Detection -** Performing clearning on $M_0$ for two types of the anomaly points: control point overflow and sign abnormality. Sending the cleaned parameters $M_1$ into the next step;
6: **for** $i = 1, 2, ...,$ to $E^{gs}$ **do**                                   ▷ **2nd Gradient Descent**
7:     $\boldsymbol{M}_1 \leftarrow \boldsymbol{M}_1 - \eta \cdot \nabla_{\boldsymbol{M}_1} L(\boldsymbol{M}_1)$;
8: **end for**
9: **Return:** The searched paramter set $\hat{\theta}$.

---

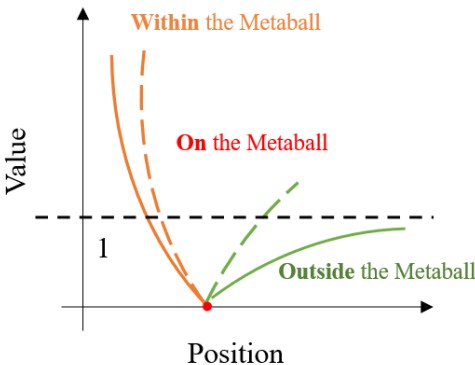

Figure 10: The value range objective function of MSE is depicted by the objective solid line in the graph, while the dashed line represents the loss function with Mean Square Error and of Eq. 12form. The latest later one increases the loss function value for points outside the point hull, improving the GS performance.

**Gradient Descent**: Gradient Descent is an optimization algorithm based on gradient information, which is readily available from the Metaball functions. For an objective function $L(M)$, its parameters can be updated iteratively to find the optimal:

$$M \leftarrow M - \eta \cdot \nabla_M L(M) \tag{11}$$

where $M$ represents the parameters for Gradient Descent, here is the parameter set of the fitted Metaball model; $\eta$ is the learning rate; $\nabla_M L(M)$ is the gradient of the $L(M)$ to the parameter $M$.

In this segment, the objective function for contour refinement is defined as a piecewise function instead of Eq. 7:

$$L(\boldsymbol{M}) = \begin{cases} \sum_{i=1}^{m} (f_{H_i}^l(\boldsymbol{M}) - 1)^2, & f_{H_i}^l \in [2, +\infty] \\ \sum_{i=1}^{m} |f_{H_i}^l(\boldsymbol{M}) - 1|, & f_{H_i}^l \in [1, 2] \\ \sum_{i=1}^{m} \left[ (f_{H_i}^l(\boldsymbol{M}) - 1)^2 + \frac{1}{f_{H_i}^l(\boldsymbol{M})} - 1 \right], & f_{H_i}^l \in [0, 1] \end{cases} \tag{12}$$

This is because many attempts have shown that a loss function in Mean Square Error(MSE) form can often result in distorted models with control points outside the targeted hull. This is related to the property of Metaball function. When the study point is internally close to or externally far from the Metaball hull, its corresponding function value will all be very small. This results in the value range of Eq. 7 as shown by lines in Figure 10, which makes the direct GS fall into the local optimal solution easily. This defect can be avoided through the implementation of Eq. 12. Under this form, the loss value of points outside and close to the Metaball hull can all be enlarged greatly (Dash lines in Figure 10). Such implementation can not only improve fitting efficiency but also adaptability to complex geometry of GS.

For gradient update, we adopt a pattern that the gradient for the whole dataset, the entire point hull H, will be calculated once for each round of iteration. This is because sufficient attention to all points on the targeted hull can endow the trained model with higher fidelity [32].

**Anomaly Detection**: Since the search of GS is strictly based on the gradient relationship, redundant control points can be merged reasonably in this process. But this also raises two other problems: control point overflow and sign abnormality. The control point overflow refers to solution with control points outside the target point hull $H$. And the sign abnormality means solution with control points of negative $k$ values. Those anomaly points will be cleared out by assigning a zero weight during this stage.

## Appendix B. The particle generation result.

All shape features of parents and clones in four datasets of Ottawa-sand and cobblestone datasets share similar distributions. In term of $\phi$, we observe deviation errors of 2.54%, 1.33%, 1.09% and

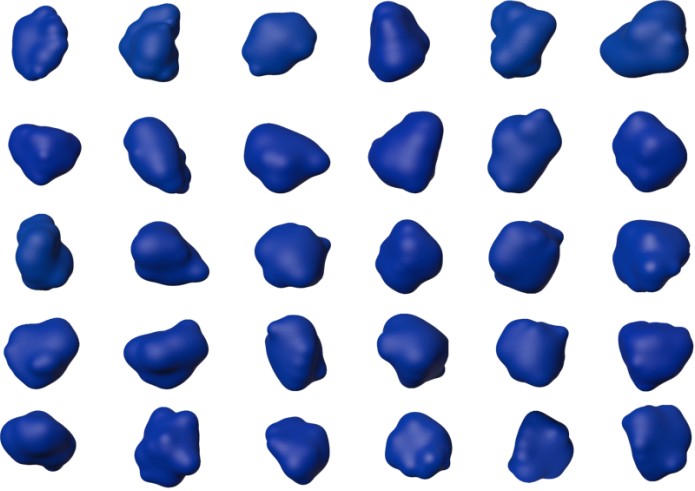

Figure 11: Examples of cloned Ottawa sands

0.92% on means between parents and clones, for the four datasets separately. In the case of standard deviation, errors of 3.12%, 5.90%, 3.77% and 4.45% are obtained. For $C$ of the Ottawa sand and cobblestone, the distribution means are off with errors of 0.03%, 0.01%, 0.10% and 0.25%. While errors in standard deviation are 2.56%, 8.32%, 4.41% and 3.46% separately. In the case of $D_{ns}$, errors of 2.57%, 0.67%, 1.29% and 0.98% are observed on the mean misalignments for the Ottawa sand and cobblestone. And the errors coming from the standard deviation are 2.69%, 1.53%, 9.35% and 5.87%. As for $CSF$, the distributions have errors of 3.87%, 5.31%, 2.80% and 3.25% on the mean, as well as errors of 11.58%, 10.70%, 1.94% and 4.88% on the standard deviations. Finally, we obtain errors of 2.77%, 6.49%, 3.72% and 0.66% from $V$, as well as errors of 4.22%, 7.45%, 3.50% and 1.41% from $A$ on means of distributions of the Ottawa sand and cobblestone. In the case of standard deviation, the errors are 20.61%, 9.87%, 1.88% and 5.64% given by $V$, as well as 24.21%, 14.94%, 6.5% and 10.65% given by $A$.

Note that the MetaballVAE is capable of learning and representing non-Gaussian shape-feature distributions. Examples of this can be seen in the histogram of the feature $V$ and $A$ of the 10 Ottawa sands and 20 cobblestone(Figure 16 and 15), where multiple peaks are observed in the distribution (although they may not be clear in the PDF curve due to limited samples). The MetaballVAE capture this feature effectively and generate particles with similar distributions, demonstrates the effectiveness of the MetaballVAE in cloning grains of complex shape-feature distributions. While an interesting phenomenon arises in the smalleset dataset of 10 Ottawa sand (Figure 15). It can be observed that the distributions of shape features of parental particles are discrete and discontinuous. Although the shape-feature distributions of the clones match well with the parents, theirs distribution are continuous with peaks around the discrete values of the parents. This is because MetaballVAE is designed to interpret and map the discrete particle morpholgoies into continuous latent space. Such architecture can not only facilitates the generation of a large variety of particles with similar shape-feature distributions and allows for more efficient exploration and manipulation of the generated particles, which will be discussed in Section 3.4.

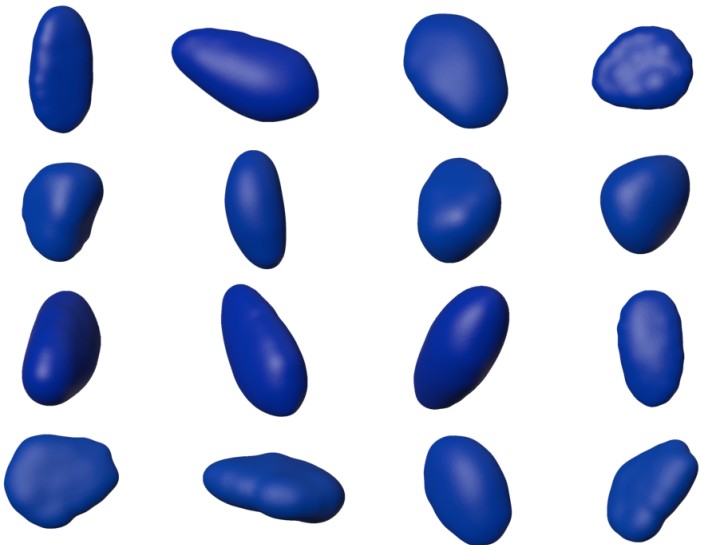

Figure 12: Examples of cloned cobblestone

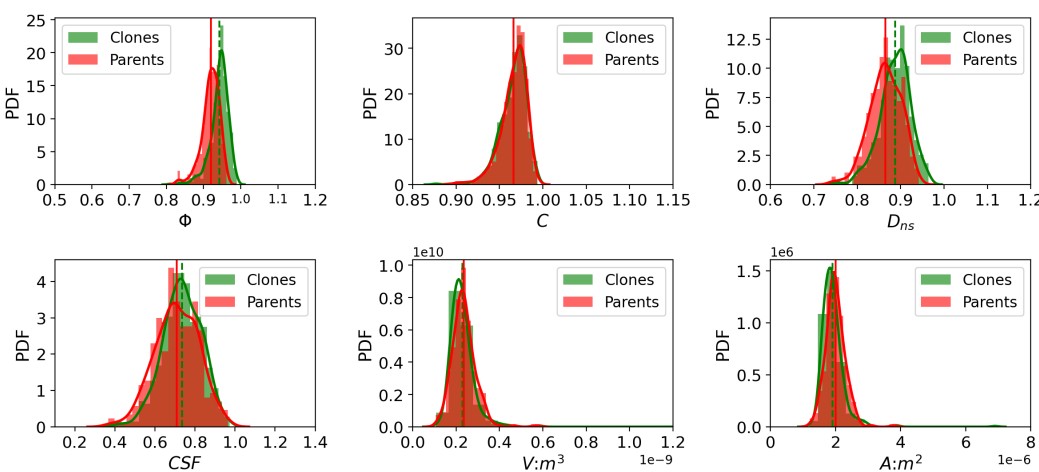

Figure 13: Comparison of feature distributions between parental and cloned particles in the Ottawa sand dataset of size 290

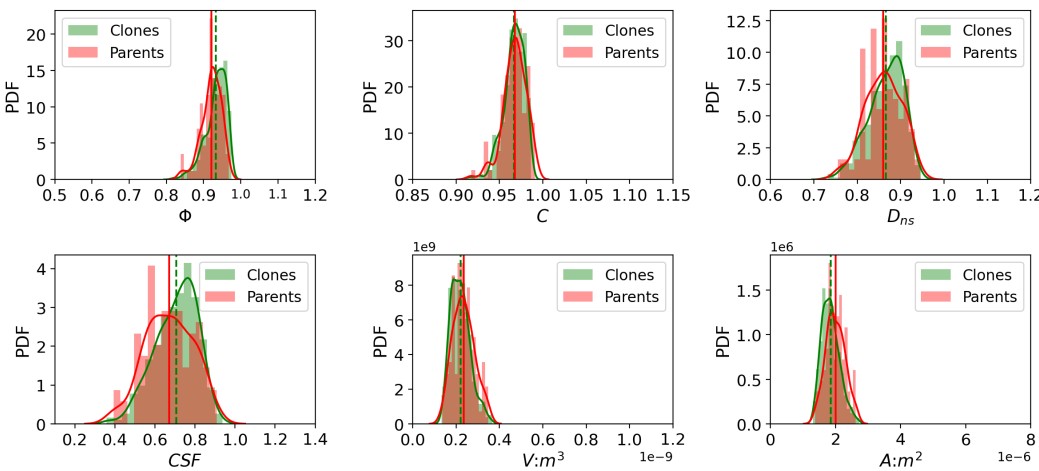

Figure 14: Comparison of feature distributions between parental and cloned particles in the Ottawa sand dataset of size 100

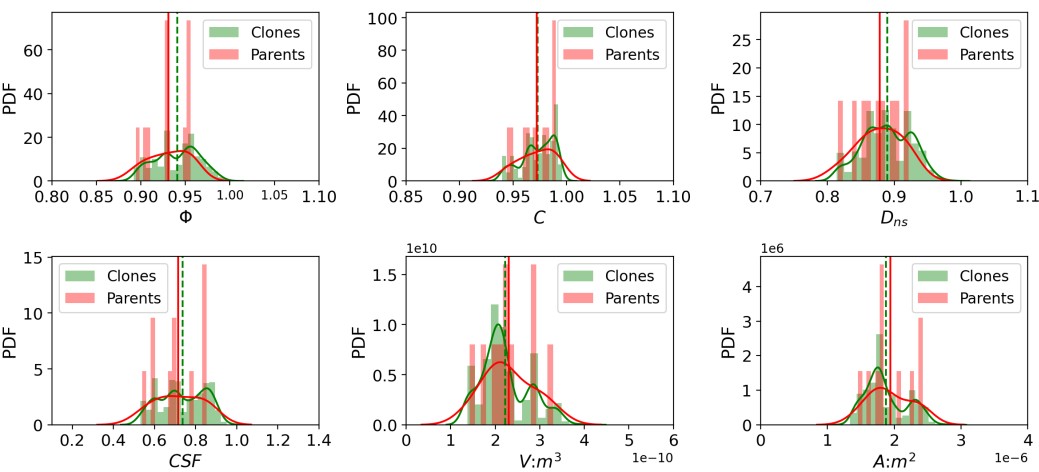

Figure 15: Comparison of feature distributions between parental and cloned particles in the Ottawa sand dataset of size 10

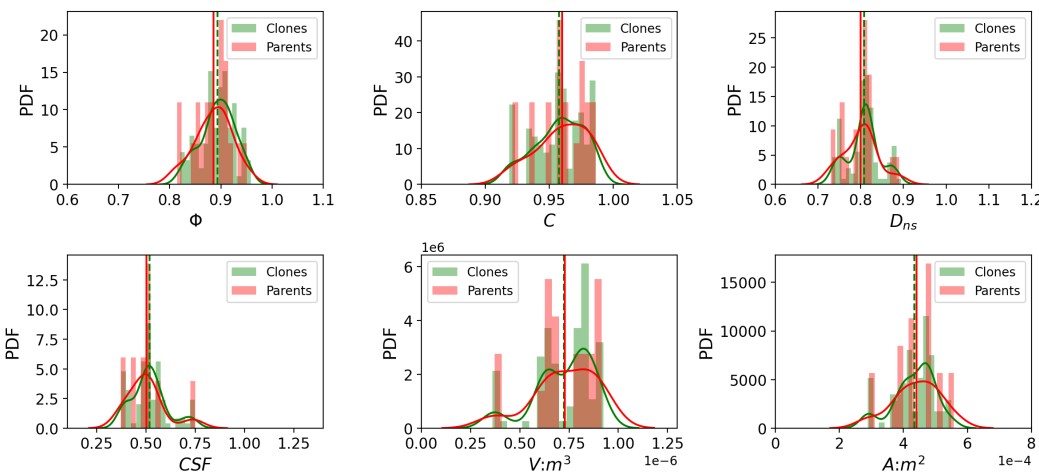

Figure 16: Comparison of feature distributions between parental and cloned particles in the cobble-stone dataset of size 20

