# OpenReview forum: "Generation of 3D Realistic Soil Particles  with Metaball Descriptor"
_NeurIPS.cc/2023/Workshop/AI4Science — NeurIPS2023-AI4Science Poster_

### Official Review · Reviewer_XCMw · 2023-10-13
**This manuscript is out of my area**

**Rating:** 6
**Confidence:** 1

**Review:**

Dear PC and ACs,

I understand this might be a bit late, but I found that this paper falls far away from my research topics. It's hard for me to evaluate it. Suggest assigning it to another reviewer.

Thank you.

---

### Official Review · Reviewer_Q3kY · 2023-10-22
**method fro 3D soil particle morphology**

**Rating:** 5
**Confidence:** 4

**Review:**

This paper proposed a method, i.e., the Metaball Variational Autoencoder for generating realistic 3D soil particle morphologies. Below are my comments.

1) The Metaball Variational Autoencoder leverages deep neural networks to generate 3D particles while preserving essential morphological features. I recommend the authors to provide a concise but more detailed explanation of how this methodology works.

2) The authors need to give details of shape control through an arithmetic pattern. They also need to explain how this enables the generation of particles with specific shapes.

3) I also wonder the ability to simulate a large number of soil particles with varying shapes and behaviors advance our understanding of soil properties and behaviors? This will underscore the practical relevance of this research.

---

### Meta-Review · Area_Chair_S9QH · 2023-10-27

**Recommendation:** Accept (Poster)
**Confidence:** 4

**Metareview:**

This paper proposes MetaballVAE, a VAE model for particle generation. This has the potential to be applied in many real science problems, e.g., molecule generation. The style-learner module proposes more promising insights of this paper. One minor comment: there has already been a lot of molecule editing / controllable generation methods, and it would be interesting to see how these two research lines overlap.